# Gibberellic Acid Inhibits *Dendrobium nobile*—*Piriformospora* Symbiosis by Regulating the Expression of Cell Wall Metabolism Genes

**DOI:** 10.3390/biom13111649

**Published:** 2023-11-14

**Authors:** Hong Chen, Yefei Li, Yuying Yin, Ji Li, Lin Li, Kunlin Wu, Lin Fang, Songjun Zeng

**Affiliations:** 1Guangdong Provincial Key Laboratory of Applied Botany, South China Botanical Garden, Chinese Academy of Sciences, Guangzhou 510650, Chinaliyefei2021@scbg.ac.cn (Y.L.); yinyuying18@mails.ucas.edu.cn (Y.Y.); liji17@scbg.ac.cn (J.L.); lilin@scib.ac.cn (L.L.); wkl8@scib.ac.cn (K.W.); 2Department of Botany, Guangzhou Institute of Forestry and Landscape Architecture, Huangzhuang South Road 6, Baiyun District, Guangzhou 510540, China; 3University of the Chinese Academy of Sciences, Beijing 100049, China; 4Key Laboratory of South China Agricultural Plant Molecular Analysis and Gene Improvement, South China Botanical Garden, Chinese Academy of Sciences, Guangzhou 510650, China

**Keywords:** protocorm, symbiosis, cell wall, hormones

## Abstract

Orchid seeds lack endosperms and depend on mycorrhizal fungi for germination and nutrition acquisition under natural conditions. *Piriformospora indica* is a mycorrhizal fungus that promotes seed germination and seedling development in epiphytic orchids, such as *Dendrobium nobile*. To understand the impact of *P. indica* on *D. nobile* seed germination, we examined endogenous hormone levels by using liquid chromatography–mass spectrometry. We performed transcriptomic analysis of *D. nobile* protocorm at two developmental stages under asymbiotic germination (AG) and symbiotic germination (SG) conditions. The result showed that the level of endogenous IAA in the SG protocorm treatments was significantly higher than that in the AG protocorm treatments. Meanwhile, GA_3_ was only detected in the SG protocorm stages. IAA and GA synthesis and signaling genes were upregulated in the SG protocorm stages. Exogenous GA_3_ application inhibited fungal colonization inside the protocorm, and a GA biosynthesis inhibitor (PAC) promoted fungal colonization. Furthermore, we found that PAC prevented fungal hyphae collapse and degeneration in the protocorm, and differentially expressed genes related to cell wall metabolism were identified between the SG and AG protocorm stages. Exogenous GA_3_ upregulated *SRC2* and *LRX4* expression, leading to decreased fungal colonization. Meanwhile, GA inhibitors upregulated *EXP6*, *EXB16*, and *EXP10-2* expression, leading to increased fungal colonization. Our findings suggest that GA regulates the expression of cell wall metabolism genes in *D. nobile*, thereby inhibiting the establishment of mycorrhizal symbiosis.

## 1. Introduction

Members of the largest flowering plant family, *Orchidaceae*, can form a complex symbiotic association with mycorrhizal fungi [1]. In addition to mineral nutrients, such as phosphorus (P) and nitrogen (N), orchid mycorrhizal fungi also supply carbohydrates for plant growth [2,3]. Orchid seed germination leads to the formation of a protocorm, a heterotrophic postembryonic structure. Fungal colonization of protocorms has three stages: a hypha enters suspensor cells of swollen embryos, pelotons form within cortical cells, and the peloton senesces and then collapses [4,5,6]. Fungal carbon (C) can be transferred from fungi to symbiotic protocorm via living or senescent hyphae. The interface between the symbionts is formed before fungal senescence, and peloton degradation also releases a significant amount of C and nitrogen in host cells [7]. The *Dendrobium* genus is one of the largest genera in the orchid family and has ornamental and medicinal value. Previous literature has reported that *Dendrobium officinale* seed germination without fungi is slower than with the mycorrhizal fungus *Tulasnella* sp. S6 [8]. *P. indica*, which is a mycorrhizal fungus with a wide range of host plants, promotes the seed germination of several species in the *Dendrobium* genus [9].

Plant hormones are essential for orchid seed germination, protocorm development, and mycorrhizal establishment [10]. Orchid symbiotic germination (SG) was enhanced by IAA, inhibited by gibberellic acid (GA), and suppressed by kinetin [11]. GA is a well-known class of phytohormones that can modulate seed germination and mycorrhizal establishment [10,12]. Bioactive GAs are formed from GGDP and catalyzed by a set of syntheses to produce GA12 [13]. GA12 is then converted to GA1/4 catalyzed by two dioxygenases (GA3ox and GA20ox), while another dioxygenase (GA2ox) deactivates bioactive GAs [14]. GA signaling proceeds via the degradation of the DELLA protein with the aid of the GA receptors GID1 and GID2, and the degradation of DELLA proteins activates downstream gene expression [15,16]. GAs were thought to act as a negative factor in AM symbiosis: GA-deficient pea mutant was thought to increase AM colonization and arbuscule incidence, and GAs were thought to modulate AM colonization via the DELLA proteins, which could promote arbuscule formation through the suppression of GA signaling [17,18]. In *D. officinale* SG, the ratio of gibberellic acids (GA_3_) and abscisic acid (ABA) was significantly higher than that during asymbiotic germination (AG); GA_3_ is a key factor connecting seed germination and mycorrhizal symbiosis, and a high concentration of exogenous GA_3_ inhibited fungal colonization and decreased the seed SG rate [10,19].

Plant growth and development are mediated by different plant hormones, and cell wall synthesis and remodeling are the terminals that respond to these signals and perform various specific functions. The cell wall is a dynamic extracellular matrix that actively controls growth and development, provides structure and support, and plays important roles in plant-microbe interactions [20]. Plant cell walls are crucial for orchid SG, and cell wall remodeling is essential for symbiotic interaction. Previous studies indicated that expansion is an extracellular protein that is involved in cell wall loosening and plant cell growth [21]. Expansins are believed to play important roles in seed germination and plant responses to biotic and abiotic stresses [22,23]. Expansins have also been identified in arbuscular mycorrhizal roots, where they play a crucial role in the accommodation process of the fungus inside the cortical cells [24]. Xyloglucan endotransglucosylase/hydrolases (XTHs) are cell wall proteins that can disrupt polysaccharide structure, promote cell wall elongation, and respond to stress [25]. Extensins are cell wall protein hydroxyproline-rich glycoproteins (HRGPs) that modify the physical and chemical characteristics of the cell wall and play an important role in the interactions between plants and microorganisms, such as symbiotic association and barrier formation against pathogens [26].

The mechanisms of plant hormones that regulate SG processes remain unclear, especially the effect of GAs on the SG process. In this study, *D. nobile* seeds were sown under AG and SG conditions. Subsequently, seed germination physiology, cytological observations of the protocorm, and the transcriptome of the protocorm development stage were assessed, and a major endogenous hormone analysis was conducted. Furthermore, exogenous GA_3_ and GA inhibitors were used to study the effect of GAs on the establishment of protocorm–symbiont interactions.

## 2. Materials and Methods

### 2.1. Seed Sample Collection

Seeds of *D. nobile* were collected from an artificial cultivation greenhouse in the South China Botanical Garden, Guangdong province, in November 2021. The collected capsules were surface sterilized for 60 s in 70% ethanol solution, followed by 20 min in 0.1% HgCl_2_ solution, and then rinsed thrice times with distilled water. Sterile seeds were stored at 4 °C in 1.5 mL sterilized tubes.

### 2.2. Seed Germination Experiments

The mycorrhizal fungus *P. indica* was grown on PDA (20 g/L potato, 20 g/L glucose, and 15 g/L agar) medium for 7 days. For SG, the seeds were suspended in 0.1% agar solution, and a single drop of the seed suspension was dispensed onto 1.5 cm × 1.5 cm filter paper on the oatmeal agar plates (OMA: 4 g/L oatmeal and 8 g/L agar) in a 9 cm Petri dish. Next, 5 mm colonies of *P. indica* on PDA medium were placed in each OMA Petri dish (containing orchid seeds) for SG. For AG, the seeds suspension was dropped onto the 1/2 MS culture medium without fungi. The conditions used for germination were a 12/12 h light/dark cycle and a temperature of 25 °C. Stage 0 (S0), Stage 1 (S1), and Stage 2 (S2) were used to determine the extent of seed germination and development. These stages represented seeds not germinated (S0), early-stage protocorm development (S1), and late-stage protocorm development (S2).

### 2.3. Histology

Protocorms from the SG, including 10, 20, and 40 days after sowing, were collected and fixed for 24 h in 50% FAA (50% ethanol, 5% acetic acid, and 3.7% formaldehyde) solution. Samples were dehydrated using a gradient of ethanol (30%, 50%, 70%, 83%, 90%, 95%, 100%); the duration of each gradient was 30 min. Next, the samples were transparentized via a gradient of xylene (30%, 50%, 70%, 100%); the duration of each gradient was 30 min. Finally, the samples were wardipped for 24 h and embedded with paraffin. The microstructural changes in the protocorm were analyzed using paraffin-embedded sections (4 μm thick) stained with 0.5% toluidine blue for 30 s. Images were captured using a LEICA DVM6 stereomicroscope. 

### 2.4. Measurement of Phytohormones

The endogenous plant hormone content of four samples corresponded to two different germination stages (S1 and S2) of symbiotic and asymbiotic seeds, respectively, and the free-living mycelium of mycorrhizal fungus and ungerminated seeds (S0) were measured using high-performance liquid chromatography–tandem mass spectrometry (HPLC–MS/MS). Samples (50 mg) were collected, ground into powder, and dissolved in 1 mL of extraction buffer (methanol/formic/water acid = 15:1:4). The extract was mixed with internal standards, and the samples were dried and dissolved with 80% methanol. Then, the organic phase was injected through a membrane filter (0.22 µm) and subsequently analyzed using an HPLC–MS/MS system equipped with a Poroshell 120 SB-C18 column (Agilent, Santa Clara, CA, USA) and a triple quadrupole tandem mass spectrometer (Quattro Premier XE; Waters, MA, USA). Three biological replicates were used for each experiment.

### 2.5. RNA Extraction and RNA-Seq

Transcriptome analysis of four samples corresponded to two different germination stages (S1 and S2) of symbiotic and asymbiotic seeds, respectively. All collected samples were immediately frozen in liquid nitrogen and used for total RNA extraction. Total RNA was extracted using the Quick RNA Isolation Kit (Huayueyang Biotech Co., Ltd., Beijing, China). RNA library construction and sequencing were performed using the HiSeq4000 platform (Illumina Inc., San Diego, CA, USA) and Wuhan Metware Biotechnology Co., Ltd. (Wuhan, China). Three biological replicate sequencing libraries were used for each stage, and the results were represented as the average value. The clean data of all samples were attained at 6 GB (Appendix A). We constructed paired-end libraries of 350 bp inserted size, PacBio 20 kb insertion libraries, and 10 × Genomics libraries. The original transcriptomic data was deposited in the public NCBI and SRA database (accession No. PRJNA943365).

The cleaned reads of *D. nobile* from the SG and AG libraries were then mapped to the *D. nobile* genome database (Appendix A). Principal component analysis (PCA) of all samples was performed during the different protocorm developmental stages. Differential expression of genes (DEGs) between samples was analyzed. Genes were considered differentially expressed if the log2 (Fold Change) ≥ 1.0 and the false discovery rate (FDR) < 0.05. To understand the DEGs in different stages between AG and SG, four comparison groups were designed: seeds grown under SG condition for 20 days vs. seeds grown under AG condition for 20 days (S-S1 vs. A-S1), seeds grown under SG condition for 40 days vs. seeds grown under AG condition for 40 days (S-S2 vs. A-S2), S-S1 vs. S-S2, and A-S1 vs. A-S2. KEGG pathway enrichment was performed to identify significantly enriched metabolic pathways.

### 2.6. Exogenous Hormone Treatment

Five-millimeter colonies of *P. indica* were inoculated onto an OMA medium with exogenous hormone GA_3_ and GAs biosynthesis inhibitor PAC; at the same time, seeds were sown onto the medium. Seeds were sown onto the OMA medium with 5 mm colonies of *P. indica* as the control, sown after 28 days; protocorms were used for phenotype observation and qPCR analysis. Meanwhile, protocorms were fixed for 24 h in 50% FAA solution for histology observation. The percentage of protocorm were calculated by protocorms/total seeds.

### 2.7. Quantitative Real-Time PCR

Total RNA was extracted from protocorm, and the first strand cDNAs were generated from equal amounts of total RNA by using a reverse transcription kit (TaKaRa, Shiga, Japan). Quantitative reverse transcription PCR (qRT-PCR) was performed with three replicates using a Roche Light Cycler 480 real-time PCR machine with SYBR Premix ExTaq II (TaKaRa). PCRs were carried at 95 °C for 30 s, followed by 40 cycles of 95 °C for 5 s, 60 °C for 30 s, and then 60 °C for 15 s. The *Dnactin* gene was used as the internal control, and relative expression levels of genes were calculated using the 2^−∆∆CT^ method. Primer sequences are listed in Appendix A.

### 2.8. Statistical Analysis

Statistical analysis was performed using Student’s *t*-test (*p* < 0.05), as indicated in the Figure legends, and all experiments were performed on at least three independently grown biological replicates. All values represent the mean ± SD. The data on hormone content were analyzed using one-way ANOVA. Duncan post hoc test was applied in the statistical analysis by using SPSS 17.0 software.

## 3. Results

### 3.1. Morphological Characterization of AG and SG D. nobile Seeds

Orchid seed germination usually begins with the emergence of an embryo from the testa and is followed by protocorm formation, which is a critical stage in orchid seed development. We found that *D. nobile* seed germination under SG conditions was faster than *D. nobile* seed germination under AG conditions. At 10 d, most embryos swelled and enlarged, and some embryos had emerged from the seed coat in both the SG and AG (Figure 1A,a). At 20 days, most embryos had emerged from the seed coat; 81.88% of the seeds formed the protocorm under SG conditions, and 50.65% formed the protocorm under AG conditions (Figure 1D). The sizes of the protocorms formed under SG conditions were larger than those formed under AG conditions (Figure 1B,b). At 40 days, protocorms exhibiting initiation of leaf differentiation and shoot appearance appeared under the SG and AG conditions; however, the protocorms under SG conditions were greener than those under AG (Figure 1C,c). The percentage of protocorms formed in the SG condition was significantly higher than that formed in the AG condition at 20 days, and the developmental process of protocorms in the SG condition was faster than that in the AG condition at 40 days. 

Histological observations of the *D. nobile* germination stages in SG were performed using paraffin-embedded sectioning. After 10 days, the embryos had enlarged due to the water uptake, and *P. indica* did not infect the embryos (Figure 2A). After 20 days, the embryos continued to enlarge; at the apex of embryos, cells began to divide to generate a zone of meristematic cells and form protocorms, and the large and basal cells of protocorms were colonized by *P. indica* hyphae (Figure 2B). After 40 days, further development of the protocorms was observed with the emergence of a developing shoot pole. *P. indica* hyphae primarily colonized the basal part of the protocorm and formed pelotons. These pelotons subsequently collapsed and degenerated in the basal cells (Figure 2C). Most notably, *P. indica* colonized in the basal cells of the protocorm and was not present in the meristematic cells and developing shoot poles.

### 3.2. Differences in Concentrations in Endogenous Hormones or Precursors during Protocorm Development in AG and SG Conditions

We quantified types of endogenous and their precursors, ABA, GAs, 1-ACC, and IAA, at different developmental stages in the SG and AG groups. These results showed that the S0 stage (ungerminated seeds) had the highest ABA, GAs, and IAA contents, while the ABA content decreased as seed germination progressed in both SG and AG conditions (Table 1). Notably, as seed germination progressed, the IAA contents of the S1 stage (20 days after sowing) and the S2 stage (40 days after sowing) were higher in the SG group than in the AG group. In SG, the IAA content of S1 was higher than that of S2, indicating that IAA was essential for colonization. Additionally, GA_3_ was detected in both S1 (0.0526 ng/g·FW) and S2 (2.2754 ng/g·FW) of the SG group but was not detected in S1 and S2 of the AG group. The ungerminated seeds contained GA_1_ (41.34 ng/g·FW). Furthermore, ACC content increased as seed germination progressed in AG, and in SG, ACC content declined at the S1 stage but increased at the S2 stage. In free-living mycorrhizal fungus *P. indica*, ABA, IAA, and ACC were detected, and GAs were not detected. In free-living *P. indica* and protocorm under the AG condition, GA_3_ was not detected; however, GA_3_ was detected in protocorm symbiosis with *P. indica*.

### 3.3. Transcriptomic Analysis of D. nobile Protocorm Development under AG and SG Conditions

This study focused on gene expression profiling in the protocorm stages of AG and SG. Transcriptome details of each sample are presented in Appendix A. The percentage of clean reads among the raw data (Q30) ranged from 93.71% to 94.79%, indicating high-quality reads for further analysis. The cleaned reads of *D. nobile* from the SG and AG libraries were then mapped to the *D. nobile* genome database with 75.80–84.35% and 84.77–88.09% of the reads in the SG and AG libraries, respectively (Appendix A). A PCA of all samples was performed during the different protocorm developmental stages (Appendix A). To study the mechanism of protocorm development in AG and SG at the molecular level, we analyzed the DEGs between samples. In total, 16,276 DEGs were identified among all the libraries. The number of DEGs between S-S2 and A-S2 was the highest: 2807 downregulated genes and 3127 upregulated genes (Figure 3A). A total of 269 common DEGs were identified among four comparisons, indicating that these DEGs might be highly associated with protocorm symbiosis (Figure 3B). The most enriched KEGG pathways of DEGs between S-S1 and A-S1 were the biosynthesis of metabolic pathways, biosynthesis of secondary metabolites, and plant–pathogen interactions (Figure 3C). For DEGs between S-S2 and A-S2, biosynthesis of secondary metabolites, metabolic pathways, and plant hormone signal transduction were the most enriched KEGG pathways (Figure 3D), suggesting that *D. nobile* protocorm growth development stimulated by fungal association is involved in metabolic pathways, plant hormone signal transduction, and plant–pathogen interaction. The KEGG pathways of DEGs between S-S1 and S-S2 and between A-S1 and A-S2 suggested that *D. nobile* protocorm development association involves plant hormone signal transduction and metabolic pathways (Appendix A).

### 3.4. DEGs Related to Hormone Metabolism and Signal Transduction

Twelve DEGs were involved in GA biosynthesis; all were upregulated in S-S1 and S-S2, including *CPS1*, *KO2*, *KAO*, four *GA20ox* genes, one *GA3ox* gene, and four *GA2ox* genes (Figure 4A, Appendix A). We propose that mycorrhizal fungal *P. indica* symbiosis with *D. nobile* protocorm can promote the expression of GA biosynthesis genes to generate active GAs. The deactivated GA gene *GA2ox* was also upregulated to maintain GA homeostasis. The GA signaling-related gene *GID2* and two *DELLA* genes were upregulated in the S-S1 and S-S2 groups (Figure 4A, Appendix A). These results indicate that the mycorrhizal fungus *P. indica* regulates the expression of genes related to GAs synthesis and signaling pathways. We compared S-S1 with S-S2 DEGs and found that GA synthesis genes were downregulated and GA signaling genes were upregulated in S-S1, suggesting that GA signaling-related genes regulate protocorm leaf differentiation.

Auxin regulates plant growth and development, and plants maintain auxin homeostasis by regulating the synthesis, degradation, and transport of IAA [27]. IAA can be formed from tryptophan (Trp), catalyzed by flavin monooxygenase YUCCA and acylamidohydrolases AMI in Trp-dependent IAA biosynthesis pathways [28,29,30,31]. We found that six DEGs were involved in Trp-dependent IAA biosynthesis pathways: four YUCC genes and two AMI genes were upregulated in the S-S1 and S-S2 groups, respectively (Figure 4B, Appendix A). Free IAA can be transported to the action site by the auxin efflux transporter PIN-formed (PIN) [32,33]. Free IAA is oxidized into 2-oxoindole-3-acetic acid (oxIAA) by the DIOXYGENASE OF AUXIN OXIDATION (DAO) enzyme [34,35]. Seven DEGs were identified: five DAO genes were upregulated, and two PIN genes were downregulated in S-S1 and S-S2, respectively (Figure 4B, Appendix A). Aux/IAA, ARFs, and SAURs are involved in the auxin signaling pathway to regulate plant growth and development [36]. In S-S1 and S-S2, we found that 12 auxin-responsive DEGs, three IAA genes, and six SAURs genes were upregulated and that three ARFs genes were downregulated (Figure 4B and Appendix A). These results indicate that the mycorrhizal fungal *P. indica* regulates the expression of genes related to IAA synthesis, degradation, and signaling pathways.

### 3.5. Effect of Exogenous GA_3_ and GA Biosynthesis Inhibitor (PAC) Treatment on Protocorm-Symbiosis Establishment

To confirm whether the GA influences the symbioses establishment between *D. nobile* and *P. indica* during SG, we performed exogenous hormone treatment experiments. Concentrations of 0.1 μM GA_3_ and 1 μM PAC were added exogenously in SG medium with *P. indica* without an exogenous hormone as a control. After 28 days of sowing, seed germination progress was slightly inhibited by 0.1 μM GA_3_, but seedlings reached the leaf differentiation stage. However, germination was hindered in the presence of 1 μM PAC, with seed germination arresting at the protocorm stage in comparison with the control (Figure 5A–C). Nonetheless, the percentage of protocorm formed in GA_3_ and PAC treatment was lower than that in the control treatment (Figure 5G). Protocorm formation was inhibited by 0.1 μM GA_3_ and 1 μM PAC. Additionally, there were fewer fungal peloton hyphae colonized in the protocorm in the 0.1 µM GA_3_ treatment (Figure 5E), and there were more fungal hyphae pelotons in the protocorm in the 1 μM PAC treatment (Figure 5F). The number of hyphae pelotons in PAC-treated protocorms was significantly higher than that in the control protocorms, and the number of hyphae pelotons in GA_3_-treated protocorms was lower than that in the control protocorms (Figure 5H). These findings reveal that PAC facilitated colonization, and GA_3_ impeded it. 

### 3.6. DEGs Related to Cell Wall Metabolism

Thirteen DEGs related to cell wall metabolism were identified between protocorms in AG and SG, including expansin genes, xyloglucan endotransglucosylase/hydrolases (XTH) genes, and extensin genes. Seven expansin genes were downregulated in S-S1 and S-S2, and three *XTH* genes and three genes encoding the extensin proteins *LRX4* and *SRC2* were upregulated in S-S1 and S-S2 (Figure 6A, Appendix A). Thus, *P. indica* colonization can regulate the expression levels of XTH, extensins, and expansin genes to modify cell wall metabolism. qPCR indicated that PAC induced *EXP6*, *EXB16* and *EXP10-2* expression and inhibited *SRC2* and *LRX4* expression, and GA_3_ induced *SRC2* and *LRX4* expression and inhibited *EXP6*, *EXB16* and *EXP10-2* expression (Figure 6B).

### 3.7. Experimental Validation of DEGs by qRT-PCR

To confirm the accuracy and reproducibility of the RNA-Seq results, we selected 12 genes related to cell wall metabolism and biosynthesis of GAs and IAA for qPCR analysis. cDNA samples were obtained from the two protocorm stages under the AG and SG conditions. The expression levels of the expansin genes *EXPA6*, *EXB16*, and *EXP10* were all lower in S-S1 and S-S2, while the gene encoding XTH and extensin genes *XTH2*, *XTH32*, *XTH22*, *LRX4*, and *SRC2* were significantly higher in S-S1 and S-S2. In S-S1 and S-S2, the genes *DAO*, *GA2ox1*, and *GA20ox1* for IAA and GAs metabolism were upregulated, which is consistent with the RNA-Seq results (Figure 7).

## 4. Discussion

### 4.1. P. indica Promotes Seed Germination and Protocorm Development of D. nobile

Orchid seed symbiotic germination was reported to be faster than asymbiotic germination [37]. We observed that *D. nobile* seeds incubated with *P. indica* germinated faster than seeds under asymbiotic germination conditions (Figure 1). During interactions between fungi and orchid seeds, fungal hyphae grow into orchid tissues through the suspensor at the base of the seeds [37]. In orchid mycorrhiza, intracellular hyphal pelotons are produced by fungal hyphae, which are eventually digested by orchid cells [37,38]. At the early stage of *D. nobile* protocorm development, hyphae pelotons formed inside the basal cells; as the protocorm further developed, the hyphae pelotons collapsed and degenerated and may finally digested by the orchid (Figure 2).

### 4.2. P. indica Colonization Can Modify the GA and IAA Metabolism-Related Expression in D. nobile

In soybeans, inoculation with compatible rhizobia can induce GA biosynthesis gene expression, and transcriptome studies using tomato and *Medicago truncatula* have demonstrated that several GA biosynthesis genes were upregulated upon inoculation with mycorrhizal fungi [39,40]. The metabolites of the bacteria and/or mycorrhizal fungus can induce the level of endogenous hormones in the orchid [11]. The GA synthesis genes *CPS1*, *KO2*, *KAO*, four *GA20ox* genes, and one *GA3ox* gene were upregulated to promote protocorm synthesis GA_3_ (Figure 4A). Exogenous GA negatively affected seed germination and fungal colonization, and endogenous bioactive GA was actively converted to inactive form during seed germination [10]. Orchids have evolved such that they do not use bioactive GA as a positive regulator of seed germination; instead, orchids auto-activate the mycorrhizal symbiosis pathway through GA inactivation to accept the fungal partner immediately during seed germination [41]. Deactivation of bioactive GAs and their precursors is catalyzed by another dioxygenase, GA 2-oxidase (GA2ox), and the expression level of *GA2ox* is upregulated in S-S1 and S-S2 when compared with that in A-S1 and A-S2 (Figure 4), indicating that orchid through GA inactivation to accept the fungal.

During arbuscular mycorrhizal symbiosis, IAA plays a positive role in root mycorrhizal colonization by controlling strigolactone levels [42]. Auxin signaling within host roots is required for the early stages of arbuscular mycorrhizal formation, including presymbiotic signal exchange [43,44]. When *D. officinale* seeds were inoculated with the mycorrhizal fungus, the rate of embryo differentiation rose, and IAA production increased significantly [10]. Orchid-associated bacteria can produce IAA to enhance symbiotic germination [11]. α-naph-thaleneacetic acid (NAA) or 6-benzylaminopurine (BA) can increase the *Calanthe discolor* protocorm formation [45]. During seed germination development, *P. indica* can upregulate the IAA synthesis genes to increase the level of IAA contents in protocorms, with the aim of accelerating the development of protocorms (Table 1, Figure 1 and Figure 4). 

### 4.3. P. indica Colonization Can Induce Cell Wall Remodeling Gene Expression

Plants need a “reprogram cycle” to adjust growth and development during a symbiotic process. The mechanism of cell wall penetration by mycorrhizal fungi involves cell wall synthesis, remodeling, and softening genes [46]. Expansins, plant cell wall loosening proteins, have been implicated in arbuscular mycorrhiza fungus (AMF) symbiosis [24,47]. For instance, the downregulation of the expansin-like gene *EXPLB1* was associated with reduced AMF infection in tomatoes [48]. α-expansin NtEXPA5 was found in mycorrhizal-infected tobacco roots, suggesting the involvement of expansin in root symbiosis [49]. Transcriptome analysis of arbuscular mycorrhizal roots showed that expansin-like genes were upregulated in epidermal cells during the formation of the pre-penetration apparatus in *M. truncatula* [50]. In soybeans, the expression of several β-EXP genes is associated with mycorrhizal colonization [51]. In our study, seven expansin genes were downregulated in S-S1 and S-S2, implying their involvement in the establishment of protocorm symbiosis (Figure 6). Extensins are cell wall proteins belonging to the superfamily of HRGPs, which play an important role in establishing a symbiotic relationship. In *D. officinale* protocorm, HRGPs accumulate in response to colonization by fungi [37]. *LRX4* and *SRC2*, which encode extensin proteins, were upregulated in SG (Figure 6A). This suggests that *P. indica* colonization regulates the expression of cell wall metabolism genes, including expansins and extensins. Furthermore, GA is an upstream regulator of the cell wall remodeling process that inhibits orchid mycorrhizal symbiosis, while GA_3_ can upregulate *SRC2* and *LRX4* expression and downregulated *EXP6*, *EXB16* and *EXP10-2* expression. As the concentration of GA_3_ and PAC increased, the inhibition of protocorm development became more obvious (Figure 6 and Appendix A). When the concentration of GA_3_ increased, the expression levels of *SRC2* and *LRX4* increased and those of *EXP6*, *EXB16*, and *EXP10-2* decreased. As the concentration of PAC increased, the expression levels of *SRC2* and *LRX4* reduced and those of *EXP6*, *EXB16*, and *EXP10-2* increased (Figure 6B).

### 4.4. GA Disrupts the Balance between Fungal Colonization and Protocorm Development in Protocorm-Symbiont Establishment

Ferguson et al. found that the GA biosynthesis mutants of *Pisum sativum* develop significantly fewer nodules than their wild-type parents [52,53]; treatment with GA biosynthesis inhibitors before rhizobial inoculation hindered the lateral root-base nodule formation in the semi-aquatic legume *Sesbania rostrata* [40]. In AMF, exogenous GA_3_ suppressed arbuscule formation in pea roots, while endogenous GA levels and GA signaling influenced the formation of arbuscules in mycorrhizal pea roots [54]. Meanwhile, GA exerted no stimulatory effect on seed germination and protocorm formation in *Calanthe discolor* asymbiotic germination [45]. However, GA played a negative role in *D. officinale* symbiotic germination and inhibited fungal colonization [45]. Exogenous GA_3_ inhibited *P. indica* colonization, while the GA biosynthesis inhibitor PAC promoted *P. indica* colonization (Figure 5F–H). We found that fewer fungal peloton hyphae colonized the protocorm in the 0.1 µM GA_3_ treatment (Figure 5E), and there were more fungal hyphae and hyphae that could not form pelotons to collapse and degenerate in the protocorm in the 1 μM PAC treatment (Figure 5F). GA not only inhibits AM symbiosis but also nodulation [55]. Furthermore, GA is an upstream regulator of the cell wall remodeling process, while GA_3_ inhibits orchid mycorrhizal symbiosis establishment. Meanwhile, GA inhibitors promote increased fungal colonization but inhibit protocorm growth, presumably due to a reduction in the rate of pelotone degradation, thereby reducing the nutrient supply to developing protocorms (Figure 8).

## 5. Conclusions

Our study found that *P. indica* could promote *D. nobile* seed germination and protocorm formation. *P. indica* colonization is associated with the regulation of GAs and IAA synthesis and signaling gene expression. Specifically, GA inhibits fungal colonization, and GA biosynthesis inhibitors promote fungal colonization. GA can inhibit the establishment of symbiosis by regulating cell wall loosen-related gene expression, while GA biosynthesis inhibitors can promote symbiosis establishment by upregulating cell wall loosen-related gene expression. Excessive or insufficient GA disrupts the balance between fungal colonization and protocorm development in the protocorm–symbiont establishment.

## Figures and Tables

**Figure 1 biomolecules-13-01649-f001:**
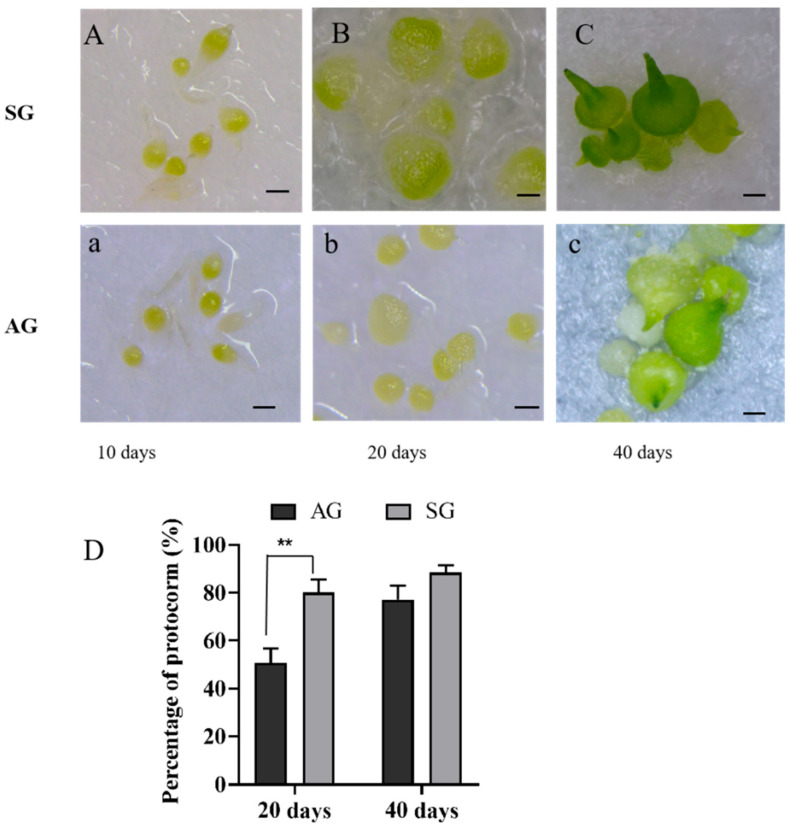
Morphological characters and seed germination stages of *D. nobile* under symbiotic germination (SG) and asymbiotic germination (AG) conditions. (**A**–**C**): symbiotic germination on the OMA with mycorrhizal fungi *P. indica*, (**a**–**c**): asymbotic germination. (**A**,**a**): Ten days after sowing seeds: embryo swollen, turned light green, rupture of testa; (**B**,**b**): Twenty days after sowing seeds: the appearance of protomeristem (protocorm stage1); (**C**,**c**): Forty days after sowing seeds: leaf differentiation, shoot appearance in protocorm (protocorm stage 2); scale bar = 100 µm. (**D**): Effect on protocorm formation of symbiotic and asymbotic treatments *t*-test (two-tailed), **, *p* < 0.01.

**Figure 2 biomolecules-13-01649-f002:**
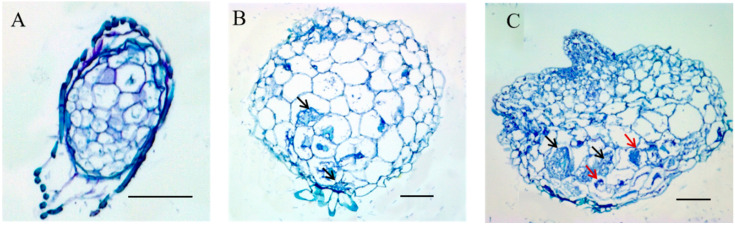
Histology of germination *D. nobile* seeds under symbiotic germination (SG) condition. (**A**) embryo swollen, the uncolonized embryo covered by testa; (**B**) protocorm with the appearance of protomeristem; (**C**) protocorm with a developing shoot pole; scale bar = 50 µm. Black arrows: peloton hyphae; red arrows: collapsing and degenerated peloton hyphae.

**Figure 3 biomolecules-13-01649-f003:**
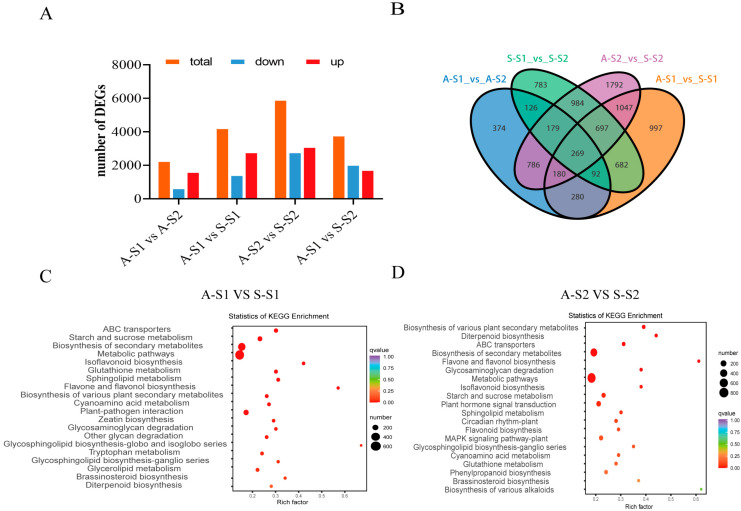
Overview of DEGs analysis of samples at different stages. (**A**) The number of up/down-regulated DEGs in each compared group. (**B**) Venn diagram of the number of DEGs in each compared group. (**C**) The analysis of KEGG enrichment in the comparison group A-S1 vs. S-S1. (**D**) The analysis of KEGG enrichment analysis in the comparison group A-S2 vs. S-S2.

**Figure 4 biomolecules-13-01649-f004:**
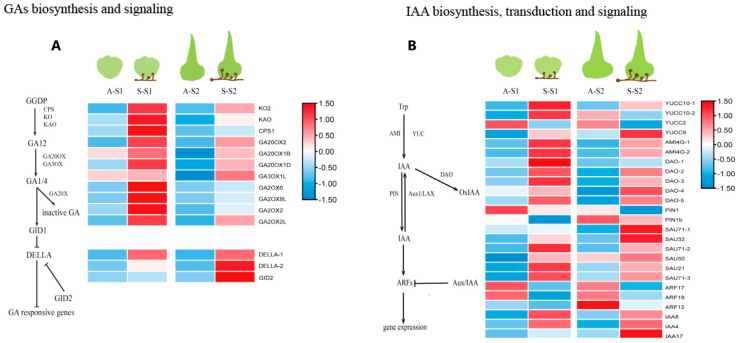
Heatmap of DEGs related to GAs and IAA metabolism and signaling. (**A**) GAs biosynthesis and signaling, (**B**) IAA biosynthesis, transduction and signaling. S-S1 and S-S2: 20 and 40 days after sowing of symbiotic germination; A-S1 and A-S2: 20 and 40 days after sowing of asymbiotic germination.

**Figure 5 biomolecules-13-01649-f005:**
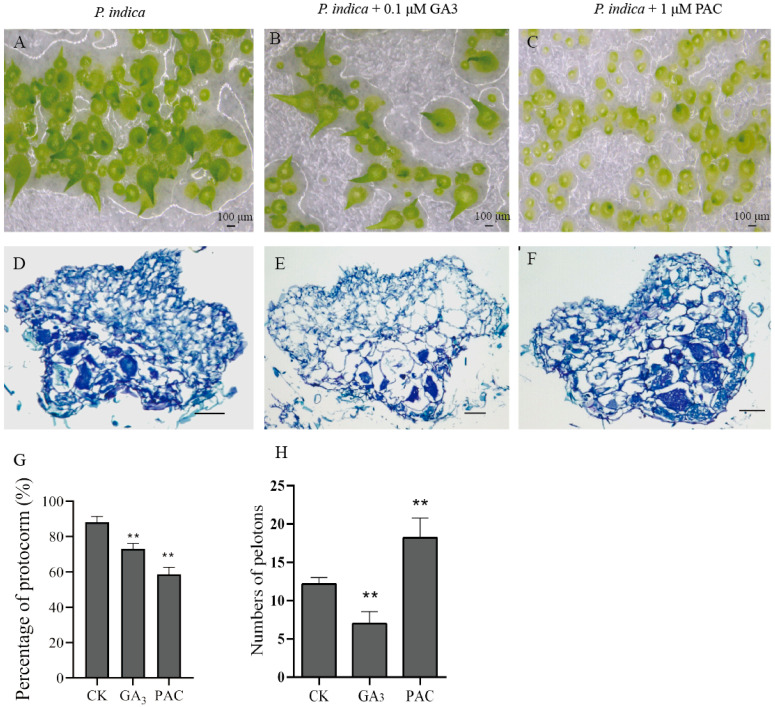
Effect of exogenous GA_3_ and PAC on *P. indica* colonization in *D. nobile* protocorm. (**A**–**C**): Morphological characters of symbiotic germination inoculated with *P. indica* at different treatments with GA_3_ and PAC. (**D**–**F**): Histology observation of protocorm inoculated with *P. indica* at different treatments with GA_3_ and PAC; scale bars = 50 µm. (**G**): Effect on protocorm formation of GA_3_ and PAC treatments, five biological replicates were analyzed; *t*-test (two-tailed). (**H**): The number of hyphae peloton in protocorm at different treatments with GA_3_ and PAC; five biological replicates were analyzed; *t*-test (two-tailed), **, *p* < 0.01.

**Figure 6 biomolecules-13-01649-f006:**
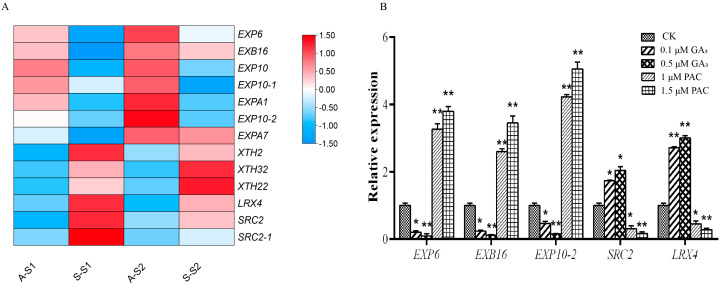
Cell wall metabolism of D. nobile protocorms under symbiotic germination (SG) and asymbiotic germination (AG) conditions (**A**): the color scale bar is shown as the log10 (FPKM + 1) value in the right; the horizontal coordinates indicate the protocorm stages under AG and SG condition. *SRC2* encoding leucine-rich repeat extensin-like protein, *LRX4* encoding LLR extensin-like protein, *EXP6* encoding expansin-A6-like, *EXB16* encoding expansin-B16-like, *EXP10-2* encoding expansin-A10-like. (**B**): Relative expression of cell wall metabolism-related genes after GA_3_ and PAC treatment. Three biological repeats were analyzed: *t*-test (two-tailed), *, *p* < 0.05; **, *p* < 0.01.

**Figure 7 biomolecules-13-01649-f007:**
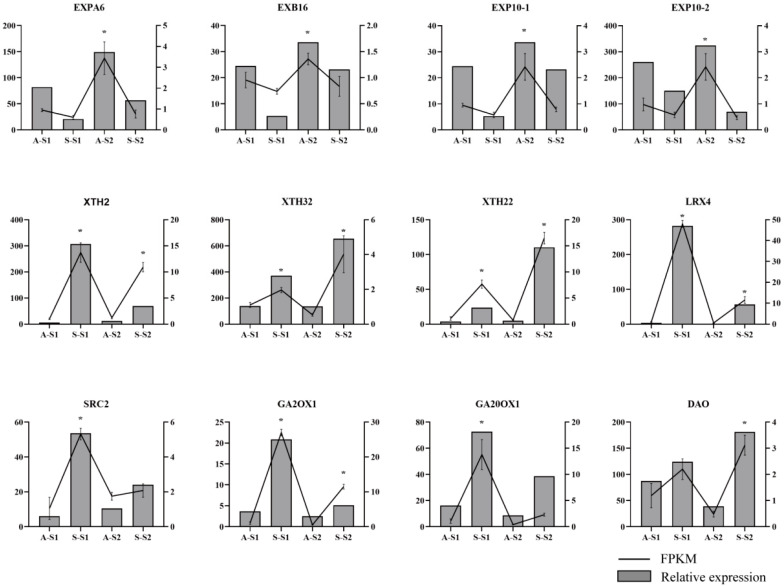
The expression levels in AG and SG protocorm development by qPCR. The reference gene was *Dnactin*. The bar chart and the left *y*-axis represent the FPKM values. The line graph and *y*-axis represent the relative expression levels by qPCR. Three biological repeats were performed: *t*-test (two-tailed), *, *p* < 0.05.

**Figure 8 biomolecules-13-01649-f008:**
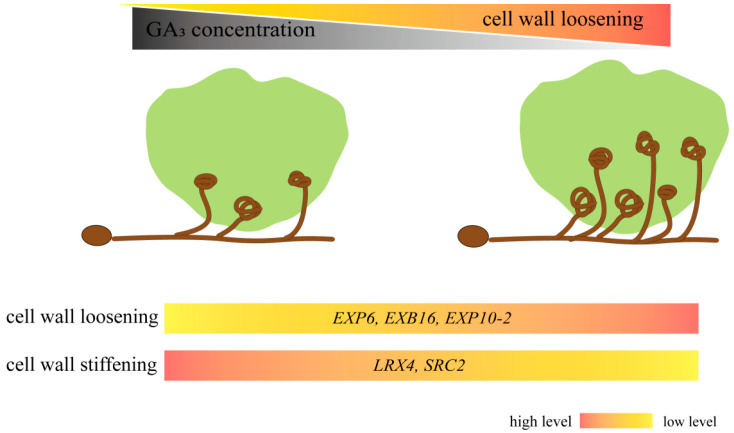
Proposed working model for the role of GA_3_ and GA inhibitor in *D. nobile*–*P. indica* symbiosis by regulating cell wall loosen-related genes expression. GA inhibited the symbiosis establishment and protocorm growth by regulating the level of cell wall metabolism gene expression. At high GA level conditions, the extensin code genes *LRX4* and *SRC2* are upregulated, and the expansin code genes *EXP6*, *EXB16*, and *EXP10-2* are downregulated, leading to cell wall stiffening, resulting in the inhibition of fungi colonization. At low GA level conditions, the extensin code genes *LRX4* and *SRC2* are downregulated, and the expansin code genes *EXP6*, *EXB16*, and *EXP10-2* are upregulated, leading to cell wall loosening, resulting in promoting fungi territory expansion and inhibiting protocorm growth.

**Table 1 biomolecules-13-01649-t001:** The level of endogenous hormones or precursor contents at different growth stages of *D. nobile* seeds, protocorms, and free-living *P. indica*.

Sample	ABA (ng/g·FW)	IAA (ng/g·FW)	GA_3_ (ng/g·FW)	GA_1_ (ng/g·FW)	ACC (ng/g·FW)
S0	29.35 ± 1.94 ^a^	14.7 ± 0.49 ^a^	N/A	41.34 ± 1.62	63.44 ± 0.95 ^d^
S-S1	1.98 ± 0.04 ^c^	14.5 ± 0.14 ^a^	0.05 ± 0.02	N/A	41.66 ± 0.30 ^e^
S-S2	1.56 ± 0.10 ^c^	8.30 ± 0.99 ^b^	2.27 ± 2.02	N/A	71.57 ± 2.00 ^c^
A-S1	2.25 ± 0.14 ^c^	3.59 ± 0.06 ^d^	N/A	N/A	85.17 ± 2.41 ^b^
A-S2	5.61 ± 0.25 ^b^	3.73 ± 0.05 ^cd^	N/A	N/A	105.9 ± 2.24 ^a^
*P. indica*	0.18 ± 0.02	2.22 ± 0.08	N/A	N/A	131.4 ± 2.52

Note: FW: fresh weight; S0: ungerminated seeds; S-S1 and S-S2: 20 days and 40 days after sowing of symbiotic germination; A-S1 and A-S2: 20 days and 40 days after sowing of asymbiotic germination; *P. indica*: free-living fungi *P. indica* on OMA. ABA: abscisic acids; IAA: indole-3-acetic acid; GA_3_ and GA_1_: Gibberellic A3 and A1; ACC: 1-Aminocyclopropanecarboxylic acid. Different letters a–e represent the significant difference in plant hormone contents at *p*-value < 0.05 with S0 as the control. Three biologic repeats were performed using Duncan method analysis in SPSS 17.0 software.

## Data Availability

All relevant data are within the paper and Appendix A.

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
