# Peer review of "Gibberellic Acid Inhibits Dendrobium nobile—Piriformospora Symbiosis by Regulating the Expression of Cell Wall Metabolism Genes"

_biomolecules, 2023, doi:10.3390/biom13111649_

Round 1
Reviewer 1 Report
Comments and Suggestions for Authors
This study aims at exploring the regulatory mechanism of orchid - Dendrobium nobile protocorm symbiosis with fungus Piriphormosphora indica, examining germination physiology, cytology, hormone analysis, and transcriptome profiling. Authors performed a wide range of analyses and I find their results interesting. Among the others – less discussed finding of lower rate of protocorm degradation under PAC treatment. This can be indirect proof that protocorm degradation is not a mere hyphae turnover (as some authors suspect) but remains under plant control. It is just a suggestion to consider by authors. In the case it will be used please elaborate more in the introduction protocorm nutrient acquisition modes (use at least https://doi.org/10.1111/nph.12700).
However, as in many cases, the manuscript has been submitter 2-3 versions too early. I would have liked the authors to have used the classic, rigorous distinction between the contents of the M&M, Results and Discussion sections, so that the reader can more easily understand the main results and conclusions drawn.
There are some deficiencies in the description of methods and insufficient discussion of the results, especially when findings contradict current knowledge in the field like f.ex the lack of gibberellins in the seeds while many studies detect these molecules at this stage ( doi: 10.1186/s40529-017-0188-4)
1. The introduction should be more explanatory leading paragraph by paragraph to the reason the study has been taken in the first place. Meanwhile,
· I advise to get rid of some unnecessary fragments like: “In Dendrobium exile ….. seedlings” or “…and was originally desert” or the whole section “Blechert et al. (1999) …. Ye, et al. 2019)” and add more general information about OM symbiosis and orchid seed germination, that are placed in the discussion. Also make sure that all the statements are supported with citation as there are many without in the current version.
· The Authors presented their hypothesis but the way the background backing the hypothesis is presented requires more clarity.
· It may stem from the fact that the literature of the topic is not enough researched and presented, including among the others https://doi.org/10.3389/fpls.2022.880600 and literature therein as regards the cell wall and (Zhao et al., 2014; Liu et al., 2015 and other positions given in the “detailed points” section of the review.
2. Materials and methods section require more details and refinements.
· Authors should define here stages of orchid seeds germination and based on this show what they mean by used terms “germination rate” and “protocorm percentage” which are otherwise confusing.
· in the section of RNA-seq (among the others: what was the length of reads, single or pair-end, we know nothing of how mapping, annotation, DEGs and enrichment analyses were performed including statistical parameters applied). Some of this information are placed in the result section making this section unnecessary lengthy.
· and statistics: t-test is only mentioned without information in what comparison it was used. Referencing for more information to figure caption violates the rule of independent description of text and figures. Moreover, there were other statistical methods or tests used f.ex. PCA as well as Duncan Test, which is a post-hoc test, so I assume ANOVA was applied and were not described and referenced properly. Also, statistical software used is described in captures (f.ex under Table 1) instead of in statistical method subchapter.
3. The results should be better discussed in confrontation with the literature on the topic under study. In many cases, I found a recapitulation of information from the results section with little attempt to critically evaluate them. Specific details are given within detailed information section of the review.
4. Moreover, there are many tipo, gramatic and syntax errors – I pointed only some of them below.
The detailed points:
Abstract:
1. Almost all orchids? No endosperm is known from orchid seeds, as no double-fertilization occurs! Cf Kodahl et al. 2015. Bot J Linn Soc 177: 202-213. Although lipids and protein (and very little starch) are stored in the embryo cells.
2. it is unnecessary to give the abbreviated name in the brackets. It is normal practice to shorten the name after its first introduction.
3. Piriformospora …seedlings” - sentence bit messy - replace for: Piriformospora indica is a mycorrhizal fungus promoting seeds germination and seedling development in epiphytic orchids - Dendrobium nobile.
4. “The level of endogenous …stages” and following sentences
In this sentence treatments rather than stages.
5. rather “lower fungal colonization” or …to “decrease of fungal colonization” (instead of less)
Results
3.1.
· “P. indica was colonized in the basal cells of the protocorm” cross out pointed words – P. indica is a coloniser not on the other way round.
· hyphae can form pelotons and provide nutrients for the protocorm through degenerating pelotons. Crossed part – too far-fetched - Authors did not make a study that can make this result legitimate to mention.
3.2
· In title – chromone level or concentration
· kinds of endogenous or precursors
3.6.
· Without proper definition of what author’s understand under “germination rate” and “percentage of protocorms” and how it was measured it is difficult to understand the result.
· “seeds reaching the leaf differentiation stage”? seedling can do this but not seeds – please be precise.
· The author tested only one and relative low concentration (in comparison with other studies) of exogenous GA3 on orchid seeds germination. Higher concentration does inhibit seeds germination to cite fragment from the introduction only “high concentration of exogenous GA3 inhibited fungal colonization and decreased the seed symbiotic germination rate (Chen et al., 2020; Wang et al., 2018).. Therefore, I recommend restraint in using the statement that “GA3 (and PAC..) do not inhibit seed germination” and in refer strictly to the concentration used especially in summary statements.
· how the statement “GA3 (…..) do not inhibit seed germination” connects with the title “GA inhibits Dendrobium nobile - Piriformospora symbiosis …”. If experimental design included seed germination with direct presence of compatible fungal hyphae and protocorms with pelotons were noted than it was a symbiotic germination. How GA could inhibit and not inhibit symbiotic germination at the same time. Please make your points clearer.
· I think that PAC (as inhibitor of GA synthesis) is simply a negative control of the GA3 application test. If so, you may simplify final interpretation of the results. See https://doi.org/10.1101/2023.04.26.538379
· I admit I am baffled by the result of DEG and qPCR. Namely, GA3 induced SRC2 and LRX4 expression (making cell wall more stiffen) and inhibited EXP6, EXB16 and EXP10-2 expression (lowering level of cell wall loosening components) , leading to less fungal colonization (so hampering colonisation) but the same profile of up and down regulation is shown for symbiotically germinated seeds (fig.5). According to the fig.8 I would have expected more of a reverse regulation allowing fungal colonisation, wouldn't you?
DISCUSSION:
4.1. There is many information here that could and should be placed in the introduction. This subchapter could be much shorter and to the point. There is no new information for the field except this particular fungus – orchid species compatibility. The rest confirms what is already known so may be shortened by giving a citation instead of elaborated descriptions.
1. All orchids in natural condition must establish associations with compatible mycorrhizal fungi. The sentence requires citation as well as following one (after coils). You should also cite Fochi et al 2017 apart of Ghirardo et al. 2020)
2. What S6 refers to?
3. rate of seeds developing into protocorm stage is it “seeds germination rate”
4. digested by protocorms - better: digested by orchid protocorm cells.
4.2. The subchapter is mostly a recapitulation of the results. It would benefit with more discussion with the results of other works on orchids. The authors, especially in the first paragraph offer no discussion with papers presenting the similar data on orchids, instead they offer comparison of GA role in germination of Brassicaceae and Fabaceae plant seeds that, unlike orchids, have endosperm rich in reserves. I would recommend discussing the results at least with doi: https://doi.org/10.1101/2023.04.26.538379, https://doi.org/10.3390/ijms21176104, https://doi.org/10.1016/0304-4238(95)00813-9, https://doi.org/10.1007/BF00009292,
· 1. in the title – “in” instead of “of”
· 2. M. truncatula - give the full generic name
FIGURES
Fig.1 characteristics not characters; make space before t in “t-test”.
FIg.2. Arrows poorly visible, make space before t in “t-test”.
FIg.4 Diagrams letters and arrows should be more explicit/bigger. I would also advise to add conclusion of what the figure presents as this drawing above heat maps are not self-explanatory.
Fig. 5 – give full explanation of abreviations. Figures need to be self-explanatory without referring to the text.
Fig. 6 I do not find added value to: a, b, c figures (magnification) Bar plot D presents the results better; E – where in the text there is information that 0.5 um Ga3 and 1.5 um PAC treatment were applied and where are the results presented? This graph does not fit to the main topic of the figure 6. Consider Fig. 5 or 7 instead.
Fig.7 - It seems to be too big. Consider making more rows than columns.
TABLES
Table 1- different growth stages of D. nobile seeds and protocorms’ 29.35
Comments on the Quality of English LanguageOverall, the quality of the English language makes it possible to read the manuscript fluently, but there are many minor grammatical and syntactical errors - I have only pointed out some of them.
Author Response
We are very grateful for the constructive comments given by you. The comments have been very helpful for us to improve our manuscript. We have carefully considered the suggestions, made a massive list of changes. Here are the responses to each point (the revised parts in the text have been marker with red).
This study aims at exploring the regulatory mechanism of orchid - Dendrobium nobileprotocorm symbiosis with fungus Piriphormosphora indica, examining germination physiology, cytology, hormone analysis, and transcriptome profiling. Authors performed a wide range of analyses and I find their results interesting. Among the others – less discussed finding of lower rate of protocorm degradation under PAC treatment. This can be indirect proof that protocorm degradation is not a mere hyphae turnover (as some authors suspect) but remains under plant control. It is just a suggestion to consider by authors. In the case it will be used please elaborate more in the introduction protocorm nutrient acquisition modes (use at least https://doi.org/10.1111/nph.12700).
Response: we have add the reference in the manuscript, meanwhile introduce the proto corm nutrient acquisition.L51-L59
However, as in many cases, the manuscript has been submitter 2-3 versions too early. I would have liked the authors to have used the classic, rigorous distinction between the contents of the M&M, Results and Discussion sections, so that the reader can more easily understand the main results and conclusions drawn.
Response: We have made a massive list of changes in M&M, Results and Discussion sections.
There are some deficiencies in the description of methods and insufficient discussion of the results, especially when findings contradict current knowledge in the field like f.ex the lack of gibberellins in the seeds while many studies detect these molecules at this stage (https://doi.org/ 10.1186/s40529-017-0188-4)
Response: In the study we did not detect GA3 in A-S1 and A-S2, maybe cause the content of GA3 in this stage is too little to detected, but we detect the precursor of GAs, GA53, GA19 and GA15 in this stage, as in A-S1 and A-S2 the expression of GA biosynthesis genes is downregulated compared with S-S1 and S-S2.
|
Sample |
GA53 (ng/g.FW) |
GA19 (ng/g.FW) |
GA24 (ng/g.FW) |
GA15 (ng/g.FW) |
|
S0 |
14.39 ± 0.40 a |
166.29 ± 8.69 a |
1.09 ± 0.43 a |
N/A |
|
S-S1 |
0.78 ± 0.01 d |
13.98 ± 0.1 b |
0.13 ± 0.003 b |
0.23 ± 0.028 a |
|
S-S2 |
0.44 ± 0.01 e |
6.43 ± 0.08 c |
0.10 ± 0.008 b |
0.17 ± 0.010 b |
|
A-S1 |
1.42 ± 0.02 b |
2.49 ± 0.14 e |
N/A |
0.21 ± 0.038 a |
|
A-S2 |
1.19 ± 0.01 c |
3.84 ± 0.23 d |
N/A |
0.09 ± 0.001 c |
- The introduction should be more explanatory leading paragraph by paragraph to the reason the study has been taken in the first place. Meanwhile,
I advise to get rid of some unnecessary fragments like: “In Dendrobium exile ….. seedlings” or “…and was originally desert” or the whole section “Blechert et al. (1999) …. Ye, et al. 2019)” and add more general information about OM symbiosis and orchid seed germination, that are placed in the discussion. Also make sure that all the statements are supported with citation as there are many without in the current version.
The Authors presented their hypothesis but the way the background backing the hypothesis is presented requires more clarity.
It may stem from the fact that the literature of the topic is not enough researched and presented, including among the others https://doi.org/10.3389/fpls.2022.880600 and literature therein as regards the cell wall and (Zhao et al., 2014; Liu et al., 2015 and other positions given in the “detailed points” section of the review.
Response: It was modified and add some background in the introduction.
- Materials and methods section require more details and refinements.
- Authors should define here stages of orchid seeds germination and based on this show what they mean by used terms “germination rate” and “protocorm percentage” which are otherwise confusing.
- in the section of RNA-seq(among the others: what was the length of reads, single or pair-end, we know nothing of how mapping, annotation, DEGs and enrichment analyses were performed including statistical parameters applied). Some of this information are placed in the result section making this section unnecessary lengthy.
- and statistics: t-test is only mentioned without information in what comparison it was used. Referencing for more information to figure caption violates the rule of independent description of text and figures. Moreover, there were other statistical methods or tests used f.ex. PCA as well as Duncan Test, which is a post-hoc test, so I assume ANOVA was applied and were not described and referenced properly. Also, statistical software used is described in captures (f.ex under Table 1) instead of in statistical method subchapter.
Response:We have made a massive list of changes in M&M
- The results should be better discussed in confrontation with the literature on the topic under study. In many cases, I found a recapitulation of information from the results section with little attempt to critically evaluate them. Specific details are given within detailed information section of the review.
Response: We added some discussed in confrontation with the literature on the topic under study in Discussion.
- Moreover, there are many tipo, gramatic and syntax errors – I pointed only some of them below.
Response: we have amended the tipo, gramatic and syntax errors which were pointed below,meanwhile modify the errors we can find.
The detailed points:
Abstract:
- Almost all orchids? No endosperm is known from orchid seeds, as no double-fertilization occurs! Cf Kodahl et al. 2015. Bot J Linn Soc 177: 202-213. Although lipids and protein (and very little starch) are stored in the embryo cells.
Response: L24
- it is unnecessary to give the abbreviated name in the brackets. It is normal practice to shorten the name after its first introduction.
Response: L26
- Piriformospora …seedlings”sentence bit messy - replace for: Piriformospora indica is a mycorrhizal fungus promoting seeds germination and seedling development in epiphytic orchids - Dendrobium nobile.
Response: L26-28
- “The level of endogenous …stages” and following sentences
In this sentence treatments rather than stages.
Response: L32-33
- rather “lower fungal colonization” or …to “decrease of fungal colonization” (instead of less)
Response: L40
Results
3.1.
- “P. indica was colonized in the basal cells of the protocorm” cross out pointed words – P. indica is a coloniser not on the other way round.
- hyphae can form pelotons and provide nutrients for the protocorm through degenerating pelotons. Crossed part – too far-fetched - Authors did not make a study that can make this result legitimate to mention.
Response:It was modified in L216, “hyphae can form pelotons and provide nutrients for the protocorm through degenerating pelotons” was deleted
3.2
- In title – chromone level or concentration
- kinds of endogenous or precursors
Response: It was modified in L225,L357-358
3.6.
Without proper definition of what author’s understand under “germination rate” and “percentage of protocorms” and how it was measured it is difficult to understand the result.
Response: The germinated seeds consist of seeds emerge from the testa and protocorms, so germination rates were calculated by germinated seeds/total seeds, percentage of protocorm were calculated by protocorms/total seeds.we modify Figure5, only show the protocorm formation rate.
“seeds reaching the leaf differentiation stage”? seedling can do this but not seeds – please be precise.
Response: It was modified the.L313
The author tested only one and relative low concentration (in comparison with other studies) of exogenous GA3 on orchid seeds germination. Higher concentration does inhibit seeds germination to cite fragment from the introduction only “high concentration of exogenous GA3 inhibited fungal colonization and decreased the seed symbiotic germination rate (Chen et al., 2020; Wang et al., 2018).. Therefore, I recommend restraint in using the statement that “GA3 (and PAC..) do not inhibit seed germination” and in refer strictly to the concentration used especially in summary statements.
Response: we have modified the sentence more strictly.L311-313
how the statement “GA3 (…..) do not inhibit seed germination” connects with the title “GA inhibits Dendrobium nobile - Piriformospora symbiosis …”. If experimental design included seed germination with direct presence of compatible fungal hyphae and protocorms with pelotons were noted than it was a symbiotic germination. How GA could inhibit and not inhibit symbiotic germination at the same time. Please make your points clearer.
Response: we have modified the sentence to make the points clearer L315-318.
- I think that PAC (as inhibitor of GA synthesis) is simply a negative control of the GA3 application test. If so, you may simplify final interpretation of the results. See https://doi.org/10.1101/2023.04.26.538379
Response: The fungal colonization in PAC treatment, were opposed to GA3 treatment.but in PAC and GA3 treatment the protocorm development were all inhibited, So we think the balance between protocorm develop and symbiosis establishment need GA regulated. How to regulated need the next study, maybe comparative of the GA contents of compatible and uncompatible fungi symbiotic protocorm will give something useful.
- I admit I am baffled by the result of DEG and qPCR. Namely, GA3 induced SRC2 and LRX4 expression (making cell wall more stiffen) and inhibited EXP6, EXB16 and EXP10-2 expression (lowering level of cell wall loosening components) , leading to less fungal colonization(so hampering colonisation) but the same profile of up and down regulation is shown for symbiotically germinated seeds (fig.5). According to the fig.8 I would have expected more of a reverse regulation allowing fungal colonisation, wouldn't you?
Response: In S-S1 and S-S2 the expression level of SRC2 and LRX4 were down-regulated while GA3 induced SRC2 and LRX4 expression in S-S1 and S-S2 the expression level of EXP6. GA3 less fungal colonization, in symbiotically germinated seeds need fungal colonization compared with asymbiotically germinated seeds, the expression of genes like PAC treatment, were opposed to GA3 treatment.
DISCUSSION:
4.1. There is many information here that could and should be placed in the introduction. This subchapter could be much shorter and to the point. There is no new information for the field except this particular fungus – orchid species compatibility. The rest confirms what is already known so may be shortened by giving a citation instead of elaborated descriptions.
- Allorchids in natural condition must establish associations with compatible mycorrhizal fungi. The sentence requires citation as well as following one (after coils). You should also cite Fochi et al 2017 apart of Ghirardo et al. 2020)
Response: It was modified
- What S6 refers to?
Response: It means mycorrhizal fungus Tulasnella sp. S6
- rate of seeds developing into protocorm stage is it “seeds germination rate”
Response: rate of seeds developing into protocorm stage is not “seeds germination rate, seeds germination rate is the emergence of an embryo from the testa,which followed by protocorm formation. Usually seeds germination ratecontains seeds emerge from the testa and protocorms.
- digested by protocorms - better: digested by orchid protocorm cells.
Response: It was modified in L433
4.2. The subchapter is mostly a recapitulation of the results. It would benefit with more discussion with the results of other works on orchids. The authors, especially in the first paragraph offer no discussion with papers presenting the similar data on orchids, instead they offer comparison of GA role in germination of Brassicaceae and Fabaceae plant seeds that, unlike orchids, have endosperm rich in reserves. I would recommend discussing the results at least with doi: https://doi.org/10.1101/2023.04.26.538379, https://doi.org/10.3390/ijms21176104, https://doi.org/10.1016/0304-4238(95)00813-9, https://doi.org/10.1007/BF00009292,
Response:we have add these reference in the manuscript,
- 1. in the title – “in” instead of “of”
Response: It was modified in L435
- 2. M. truncatula - give the full generic name
Response: It was modified in L437
FIGURES
Fig.1 characteristics not characters; make space before t in “t-test”.
Response: It was modified
FIg.2. Arrows poorly visible, make space before t in “t-test”.
Response:It was modified
FIg.4 Diagrams letters and arrows should be more explicit/bigger. I would also advise to add conclusion of what the figure presents as this drawing above heat maps are not self-explanatory.
Response:It was modified
Fig. 5 – give full explanation of abreviations. Figures need to be self-explanatory without referring to the text.
Response:It was added the full explanation of abreviations,L400-405.
Fig. 6 I do not find added value to: a, b, c figures (magnification) Bar plot D presents the results better; E – where in the text there is information that 0.5 um Ga3 and 1.5 um PAC treatment were applied and where are the results presented? This graph does not fit to the main topic of the figure 6. Consider Fig. 5 or 7 instead.
Response:we modify the Fig.5 and Fig.6, meanwhile add the 0.5 μM Ga3 and 1.5 μM PAC treatment in Figure S3
Fig.7 - It seems to be too big. Consider making more rows than columns.
Response: uauslly it needs 12 or 15 genes expression levels to test the RNA-seq data.
TABLES
Table 1- different growth stages of D. nobile seeds and protocorms
Response:It was modified in L357-358.
Please see the attachment
Reviewer 2 Report
Comments and Suggestions for Authors
Authors investigated the impact of mycorrhizal fungus (Piriformospora indica) a on Dendrobium nobile seeds germination. AUthors measured plant hormones and performed transcriptomic analysis protocorm at two developmental stages under asymbiotic and symbiotic germination conditions. AUthors found that hormone-regulated cell wall metabolism is involved in the establishment of mycorrhizal symbiosis.
Major points
1. There is no clear explanation in material and Methodss ection why two stages S1 and S2 were selected for transcriptomics and S0 was omitted, whereas other experiments includes data from the three stages.
2. Manuscript needs restructurization, Results section containg huge amount of text that should be transfered to Introduction and Material and Methods section. On the other hand results are interprested in Results (examples) what is lacking in Discussion. Discussion is written like a review of literature to which obtained results are only added and therefore constitute only a minor part of a text. Discussion should try to explain the results and give them the meaning.
3. There is a lack of underlining the novelty and originality of the results
4. Conclusion is just a summary of results without any conclusion and must be rewritten.
5. Heatmaps must be provided with data from three replicates of each stage, not only one (Figure 5). Explain how the heatmaps in Figure4 were created to show one column per treatment.
6. All cited refrences must be converted to numbers to keep the journal style.
7. Figures and tables must be cited as full name for example "Figure 1". Authors use "Fig." or "Fig". Please adjust the format to the journal requirements.
8. Confusing descriptin of statistics test used. They are different in Material and Method section and different in figure captions.
9. Support this result with numbers "protocorms under SG conditions were greener than those under AG". Was the chlorophyll concentration measured?
10. Table 1 contains "N/A" fos the measurement of GAs in many samples. Please explain.
There are no line numbers in the file, what makes a difficulty in indication by me and finding by authors the place in text.
PAGE 1
- tracked change in authors name ",_Lin",
- no closing bracket "(D. nobile seeds",
- provide full names for abbreviated gens: SRC2, LRX4 expression, EXP6,EXB16 and EXP10-2,
PAGE 2
- the lack of scpace "germination(Chen et al.," and "restricted(Ye,",
- define "Ec",
- the abbreviation of ABA is two times introduced in two following sentences,
- add the main result obtained in thos study to the last paragraph of Introduction,
PAGE 3
- provide full name of FAA,
PAGE 4
- transfer to propr section this description of method "Histological observations of the D. nobile germination stages in SG were performed using paraffin-embedded sectioning.",
PAGE 6
- element of Introduction in Results, transfer or rewrite "The coordinated signaling of endogenous hormones is essential for seed germination and protocorm development (Fang et al., 2021).",
- ABA and GAs abbreviations were already explained in th text,
- element of Discussion in Results, transfer or rewrite "This result suggests that SG requires more IAA than AG does."
- FW is explained only in table,
- there in no need to provide full values with units in the text surrounding a table in which these data is included,
- add subscripts "GA3 and GA1: gibberellic acids;",
- repeated form Methods "RNA-seq analysis of D. nobile seeds sowed under AG and SG conditions was performed 20 days after sowing (S1) and 40 days after sowing (S2)."
- description of method without any description of obtained result "Principal component analysis (PCA) of all samples was performed during the different protocorm developmental stages (Fig. S1).",
PAGE 7
- explain DEGs at first use,
- element of Metthods in Results, transfer it "Genes were considered differentially expressed (DEGs) if the log2 (Fold Change) ≥1.0 and the false discovery rate (FDR) < 0.05.",
- element of Discussion in Results, authors interpretation should be transfered to Discussion and supported with literatue, then it will get its meaning:
"indicating that these DEGs might be highly associated with protocorm symbiosis (Fig. 3B)."
" suggesting that D. nobile protocorm growth development stimulated by fungal association is involved in metabolic pathways, plant hormone signal transduction, and plant-pathogen interaction."
"suggested that D. nobile protocorm development association involves plant hormone signal transduction and metabolic pathways (Fig S2 A-B)."
- increase font size of KEGG gropu names in Figure 3C and D or their resolution because they seem blurry
PAGE 8
- elements of Introduction (sometimes suitable for Discussion) in Results, transfer it of shorten to one sentence underlying the need of such analyses:
"Plant hormones are essential for protocorm development and mycorrhizal establishment (Juan Chen et al., 2020). Bioactive GAs are formed from GGDP and catalyzed by a set of syntheses to produce GA12 (Qian Ma et al., 2011). GA12 is then converted to GA1/4 catalyzed by two dioxygenases, GA3ox and GA20ox, while another dioxygenase, GA2ox, can deactivate bioactive GAs (Yamaguchi, 2008). GA signaling proceeds by degradation of the DELLA protein with the aid of the GA receptors GID1 and GID2, and the degradation of DELLA proteins activates downstream gene expression (Sasaki et al., 2003; Ueguchi-Tanaka et al., 2005)."
"Auxin is known to regulate plant growth and development, and plants maintain auxin homeostasis by regulating the synthesis, degradation, and transport of IAA (Korasick et al., 2013). IAA can be formed from tryptophan (Trp), catalyzed by flavin monooxygenase YUCCA and acylamidohydrolases AMI in Trp-dependent IAA biosynthesis pathways (Di et al., 2016; Mano et al., 2010; Mashiguchi et al., 2011; Won et al., 2011)."
"Free IAA can be transported to the action site by the auxin efflux transporter PIN-formed (PIN) (Guan et al., 2020; Petrasek et al., 2006). Free IAA is oxidized into 2- oxoindole-3-acetic acid (oxIAA) by the DIOXYGENASE OF AUXIN OXIDATION (DAO)
enzyme (Porco et al., 2016; Zhang and Peer, 2017)."
- suitable for Conclusion part "We propose that mycorrhizal fungal P. indica symbiosis with D. nobile protocorm can promote the expression of GA biosynthesis genes to generate active GAs."
- explain abbreviations YUCC, AMI,
PAGE 9
- add a comma between biosynthesis and transduction if Figure 4 top name,
lements of Introduction (sometimes suitable for Discussion) in Results, transfer it of shorten to one sentence underlying the need of such analyses "Plant cell walls are a dynamic extracellular matrix that actively control growth and development, provide structure and support, and play important roles in plant-microbe interactions (Barnes and Anderson, 2018). Previous studies indicated that expansin are extracellular proteins, which involved in cell wall-loosening and plant cell growth (Cosgrove et al., 2002). Expansins have also been identified in arbuscular mycorrhizal roots, where they play a crucial role in the accommodation process of the fungus inside the cortical cells (Balestrini et al., 2005). Extensin proteins modify the physical and chemical characteristics of cell wall to strengthen the cell wall in response to various developmental and environmental signals."
PAGE 10
- methodological descritoion, tranfer it "Concentrations of 0.1 μM GA3 and 1 μM PAC were added exogenously in SG medium with P. indica, meanwhile without exogenous hormone as control. After 28 days of sowing, seed germination progress was slightly inhibited by the 0.1 μM GA3 with seeds reaching the leaf differentiation stage."
PAGE 11
Y axis name of Fig 6d should start with capital letter and check the grammar, shouldn't it be "Number of hyphae peletons"?
PAGE 11
it's methodology, transfer it "The primers sequence were listed in Table S6."
PAGE 12
- check English "showed germinated"
- check spaces "seeds ( Li et al., 2018"
PAGE 13
-italicize M. truncatula
- check spaces "(Fig5)."
- add a year after "Ferguson et al."
- correct "pisum"
- check spaces "parents ( Ferguson"
PAGE 14
- rewrite 'inhibitors inhibited'
- add dot "(Maekawa et al., 2009) Furthermore"
Conclusion:
remember that fungi affects only dynamics/pace of germination not the final capacity expressed in percentage
Author Response
We are very grateful for the constructive comments given by you. The comments have been very helpful for us to improve our manuscript. We have carefully considered the suggestions, made a massive list of changes. Here are the responses to each point (the revised parts in the text have been marker with red).
Open Review
Comments and Suggestions for Authors
Authors investigated the impact of mycorrhizal fungus (Piriformospora indica) a on Dendrobium nobile seeds germination. AUthors measured plant hormones and performed transcriptomic analysis protocorm at two developmental stages under asymbiotic and symbiotic germination conditions. AUthors found that hormone-regulated cell wall metabolism is involved in the establishment of mycorrhizal symbiosis.
Major points
- There is no clear explanation in material and Methodss ection why two stages S1 and S2 were selected for transcriptomics and S0 was omitted, whereas other experiments includes data from the three stages.
Response: In the study we just compare the genes expression between the same stage in symbiotic and asymbiotic germination. In fact we also experiments the S0 stage, but we did not put in the article. The GA biosynthesis genes expression is upregulated as the seeds germination.
- Manuscript needs restructurization, Results section containg huge amount of text that should be transfered to Introduction and Material and Methods section. On the other hand results are interprested in Results (examples) what is lacking in Discussion. Discussion is written like a review of literature to which obtained results are only added and therefore constitute only a minor part of a text. Discussion should try to explain the results and give them the meaning.
Response: It was modified, we restructure the manuscript,according to the review comment.
- There is a lack of underlining the novelty and originality of the results
Response: GA inhibitor fungal colonization, GAs synthesis inhibitor promote fungal colonization, GA regulated cell wall metabolism genes expression to regulated symbiosis establish are novelty and originality results. We have underlining in the conclusion.
- Conclusion is just a summary of results without any conclusion and must be rewritten.
Response: It was modified
- Heatmaps must be provided with data from three replicates of each stage, not only one (Figure 5). Explain how the heatmaps in Figure4 were created to show one column per treatment.
Response: All the data used in heatmaps were three replicates, used the average to created the heatmaps with TBtools
- All cited refrences must be converted to numbers to keep the journal style.
Response:It was modified
- Figures and tables must be cited as full name for example "Figure 1". Authors use "Fig." or "Fig". Please adjust the format to the journal requirements.
Response:It was modified
- Confusing descriptin of statistics test used. They are different in Material and Method section and different in figure captions.
Response:It was modified
- Support this result with numbers "protocorms under SG conditions were greener than those under AG". Was the chlorophyll concentration measured?
Response: We did not measured the chlorophyll concentration, we just judge from the color of protocorm
- Table 1 contains "N/A" fos the measurement of GAs in many samples. Please explain.
Response: In the study we did not detect GA3 in A-S1 and A-S2, maybe cause the content of GA3 in this stage is too little to detected, but we detect the precursor of GAs, GA53, GA19 and GA15 in this stage, as in A-S1 and A-S2 the expression of GA biosynthesis genes is downregulated compared with S-S1 and S-S2.
There are no line numbers in the file, what makes a difficulty in indication by me and finding by authors the place in text.
Response: We have added line numbers in the files
PAGE 1
- tracked change in authors name ",_Lin",
- no closing bracket "(D. nobile seeds",
- provide full names for abbreviated gens: SRC2, LRX4 expression, EXP6,EXB16 and EXP10-2,
Response: It was modified
PAGE 2
- the lack of scpace "germination(Chen et al.," and "restricted(Ye,",
- define "Ec",
- the abbreviation of ABA is two times introduced in two following sentences,
- add the main result obtained in thos study to the last paragraph of Introduction,
Response: It was modified
PAGE 3
provide full name of FAA,
Response: It was modified in L32-133
PAGE 4
transfer to propr section this description of method "Histological observations of the D. nobile germination stages in SG were performed using paraffin-embedded sectioning.",
Response: It was modified in L132-136
PAGE 6
- element of Introduction in Results, transfer or rewrite "The coordinated signaling of endogenous hormones is essential for seed germination and protocorm development (Fang et al., 2021).",
- ABA and GAs abbreviations were already explained in th text,
element of Discussion in Results, transfer or rewrite "This result suggests that SG requires more IAA than AG does."
FW is explained only in table,
there in no need to provide full values with units in the text surrounding a table in which these data is included,
add subscripts "GA3 and GA1: gibberellic acids;",
repeated form Methods "RNA-seq analysis of D. nobile seeds sowed under AG and SG conditions was performed 20 days after sowing (S1) and 40 days after sowing (S2)." description of method without any description of obtained result "Principal component analysis (PCA) of all samples was performed during the different protocorm developmental stages (Fig. S1).",
Response: It was modified in L32-133
PAGE 7
explain DEGs at first use,
- element of Metthods in Results, transfer it "Genes were considered differentially expressed (DEGs) if the log2 (Fold Change) ≥1.0 and the false discovery rate (FDR) < 0.05.",
- element of Discussion in Results, authors interpretation should be transfered to Discussion and supported with literatue, then it will get its meaning:
"indicating that these DEGs might be highly associated with protocorm symbiosis (Fig. 3B)."
" suggesting that D. nobile protocorm growth development stimulated by fungal association is involved in metabolic pathways, plant hormone signal transduction, and plant-pathogen interaction."
"suggested that D. nobile protocorm development association involves plant hormone signal transduction and metabolic pathways (Fig S2 A-B)."
- increase font size of KEGG gropu names in Figure 3C and D or their resolution because they seem blurry
Response: It was modified in L32-133
PAGE 8- elements of Introduction (sometimes suitable for Discussion) in Results, transfer it of shorten to one sentence underlying the need of such analyses:
"Plant hormones are essential for protocorm development and mycorrhizal establishment (Juan Chen et al., 2020). Bioactive GAs are formed from GGDP and catalyzed by a set of syntheses to produce GA12 (Qian Ma et al., 2011). GA12 is then converted to GA1/4 catalyzed by two dioxygenases, GA3ox and GA20ox, while another dioxygenase, GA2ox, can deactivate bioactive GAs (Yamaguchi, 2008). GA signaling proceeds by degradation of the DELLA protein with the aid of the GA receptors GID1 and GID2, and the degradation of DELLA proteins activates downstream gene expression (Sasaki et al., 2003; Ueguchi-Tanaka et al., 2005)."
"Auxin is known to regulate plant growth and development, and plants maintain auxin homeostasis by regulating the synthesis, degradation, and transport of IAA (Korasick et al., 2013). IAA can be formed from tryptophan (Trp), catalyzed by flavin monooxygenase YUCCA and acylamidohydrolases AMI in Trp-dependent IAA biosynthesis pathways (Di et al., 2016; Mano et al., 2010; Mashiguchi et al., 2011; Won et al., 2011)."
"Free IAA can be transported to the action site by the auxin efflux transporter PIN-formed (PIN) (Guan et al., 2020; Petrasek et al., 2006). Free IAA is oxidized into 2- oxoindole-3-acetic acid (oxIAA) by the DIOXYGENASE OF AUXIN OXIDATION (DAO)
enzyme (Porco et al., 2016; Zhang and Peer, 2017)."
- uitable for Conclusion part "We propose that mycorrhizal fungal P. indica symbiosis with D. nobile protocorm can promote the expression of GA biosynthesis genes to generate active GAs."
- explain abbreviations YUCC, AMI,
Response: It was modified in L32-133
PAGE 9
- add a comma between biosynthesis and transduction if Figure 4 top name,
lements of Introduction (sometimes suitable for Discussion) in Results, transfer it of shorten to one sentence underlying the need of such analyses "Plant cell walls are a dynamic extracellular matrix that actively control growth and development, provide structure and support, and play important roles in plant-microbe interactions (Barnes and Anderson, 2018). Previous studies indicated that expansin are extracellular proteins, which involved in cell wall-loosening and plant cell growth (Cosgrove et al., 2002). Expansins have also been identified in arbuscular mycorrhizal roots, where they play a crucial role in the accommodation process of the fungus inside the cortical cells (Balestrini et al., 2005). Extensin proteins modify the physical and chemical characteristics of cell wall to strengthen the cell wall in response to various developmental and environmental signals."
Response:It was modified according to the review comment
PAGE 10
- methodological descritoion, tranfer it "Concentrations of 0.1 μM GA3 and 1 μM PAC were added exogenously in SG medium with P. indica, meanwhile without exogenous hormone as control. After 28 days of sowing, seed germination progress was slightly inhibited by the 0.1 μM GA3 with seeds reaching the leaf differentiation stage."
Response: It was modified in L32-133
PAGE 11
Y axis name of Fig 6d should start with capital letter and check the grammar, shouldn't it be "Number of hyphae peletons"?
Response: It means the protocorm contain hyphae peletons number
PAGE 11
it's methodology, transfer it "The primers sequence were listed in Table S6."
Response: The primers sequence were listed in Table S6 was moved in M&M
PAGE 12
- check English "showed germinated"
check spaces "seeds ( Li et al., 2018"
Response: It was modified
PAGE 13
-italicize M. truncatula
check spaces "(Fig5)."
- add a year after "Ferguson et al."
- correct "pisum"
- check spaces "parents ( Ferguson"
Response: It was modified
PAGE 14
- rewrite 'inhibitors inhibited'
- add dot "(Maekawa et al., 2009) Furthermore"
Response: It was modified
Round 2
Reviewer 1 Report
Comments and Suggestions for Authors
Dear Authors, please review the entire text carefully, as in many cases spaces are missing between text and brackets (e.g. when numbering literature, but not only).
Introduction:
· There are significant grammatical errors in the added text (incorrect tenses, singular/plural nouns, lack of capital letters, etc.) There is also a conceptual mess. Please write simply, clearly, and grammatically correctly that C can be transferred from fungi to orchids via living or dead hyphae.
· Plant hormones part should go first before the study model (Dendrobium-P.indica)
· the authors have presented the background in such a way that the reader would not know what the authors mean by 'to further investigate this process'. There is still no scientific problem signalled in the introduction - I pointed it out in a previous review.
M&M
2.5. There are still missing information I mentioned about before.
„what was the length of reads, single or pair-end, we know nothing of how enrichment analyses were performed including statistical parameters applied). Some of this information are placed in the result section making this section unnecessary lengthy”
2.6 – Homogenise font size
2. 8 – Replace he statistical analysis was using Duncan method” with „Duncan post-hoc test was applied”
Results
3.2
· Once again – in the title add what kind of difference you measured: in level or concentration?
endogenous hormons or its precursors:
3.3
· The whole section should go to M&M
To understand the DEGs in different stages between AG and SG, four com-parison groups were designed: seeds grown under SG condition for 20 days vs. Seeds grown under AG condition for 20 days (S-S1 vs A-S1), seeds grown under SG conditio for 40 days vs. seeds grown under AG condition for 40 days (S-S2 vs A-S2), S-S1 vs S-S2, and A-S1 vs A-S2
3.5
· Again - replace „seeds have germinated to the leaf differentiation stage” with „but seedling reached the leaf differentiation stage.
Discussion
4.1
Erase „showed” – it is redundant.
Digested by orchid not protocorm cels
4.3 – not concentrate but concentration
4.4
„promoted P. indica colonization (Figure 5 D-F), I think it more corect to cite Figure 5 F, H)
„GA3 inhibits orchid mycorrhizal symbiosis” rather „symbiosis establishment”
„more fungal colonization and inhibiting the protocorm growth” – Please develope the sentence more. It is not so obvious why f.ex.: GA inhibitors promote greater fungal colonisation but inhibit protocorm growth, presumably due to a reduction in the rate of pelotone degradation, thus reducing the nutrient supply to developing protocorms”
Conclusions
This sentence : As the protocorm develops, the pelotons collapse, degenerate. It is just to trivial to place in the conclusion.
FIGURES
Fig. 2 Change capture for „Histology of germination D. nobile seeds under symbiotic germination (SG) condition”
Fig. 5. Remove red squers from DEF or describe in the capture. Homogenise the letter size in G and H legends. Hyphae pelotons? It is enough to write pelotons as there are no „no-hyphae peletons”; „five biological replicates were performer” – analysed – not performer; change for subscript in GA3.
Fig. 6 Give more generalised title here f.ex „Cell wall metabolism of D. nobile protocorms under symbiotic germination (SG) and asymbiotic germination (AG) conditions” and more speific description of A and B
Comments on the Quality of English Language
The English language - especially in the corrected fragments - needs improvement.
Author Response
We are very grateful for the constructive comments given by you again. The comments have been very helpful for us to improve our manuscript. We have carefully considered the suggestions, made a massive list of changes. Here are the responses to each point (the revised parts in the text have been marker with red).
- There are significant grammatical errors in the added text (incorrect tenses, singular/plural nouns, lack of capital letters, etc.) There is also a conceptual mess. Please write simply, clearly, and grammatically correctly that C can be transferred from fungi to orchids via living or dead hyphae.
Response: L55-56
- Plant hormones part should go first before the study model (Dendrobium-P.indica)
- the authors have presented the background in such a way that the reader would not know what the authors mean by 'to further investigate this process'. There is still no scientific problem signalled in the introduction - I pointed it out in a previous review.
Response: L101
M&M
2.5. There are still missing information I mentioned about before.
„what was the length of reads, single or pair-end, we know nothing of how enrichment analyses were performed including statistical parameters applied). Some of this information are placed in the result section making this section unnecessary lengthy”
Response:L157,L165-171
2.6 – Homogenise font size
Response:It was modified
- 8 – Replace he statistical analysis was using Duncan method” with „Duncan post-hoc test was applied”
Response:L193-194
Results
3.2
- Once again – in the title add what kind of difference you measured: in level or concentration?
endogenous hormons or its precursors:
Response:L224-225
3.3
- The whole section should go to M&M
To understand the DEGs in different stages between AG and SG, four com-parison groups were designed: seeds grown under SG condition for 20 days vs. Seeds grown under AG condition for 20 days (S-S1 vs A-S1), seeds grown under SG conditio for 40 days vs. seeds grown under AG condition for 40 days (S-S2 vs A-S2), S-S1 vs S-S2, and A-S1 vs A-S2
Response: L165-169
3.5
- Again - replace „seeds have germinated to the leaf differentiation stage” with „but seedling reached the leaf differentiation stage.
Response:L307
Discussion
4.1
Erase „showed” – it is redundant.
Response:L416
Digested by orchid not protocorm cels
Response:L424
4.3 – not concentrate but concentration
Response:L476
4.4
„promoted P. indica colonization (Figure 5 D-F), I think it more corect to cite Figure 5 F, H)
Response:L495
„GA3 inhibits orchid mycorrhizal symbiosis” rather „symbiosis establishment”
Response:L501
„more fungal colonization and inhibiting the protocorm growth” – Please develope the sentence more. It is not so obvious why f.ex.: GA inhibitors promote greater fungal colonisation but inhibit protocorm growth, presumably due to a reduction in the rate of pelotone degradation, thus reducing the nutrient supply to developing protocorms”
Response:L501-504
Conclusions
This sentence : As the protocorm develops, the pelotons collapse, degenerate. It is just to trivial to place in the conclusion.
Response:It was deleted
FIGURES
Fig. 2 Change capture for „Histology of germination D. nobile seeds under symbiotic germination (SG) condition”
Response:L365-366
Fig. 5. Remove red squers from DEF or describe in the capture. Homogenise the letter size in G and H legends. Hyphae pelotons? It is enough to write pelotons as there are no „no-hyphae peletons”; „five biological replicates were performer” – analysed – not performer; change for subscript in GA3.
Response:L381-389
Fig. 6 Give more generalised title here f.ex „Cell wall metabolism of D. nobile protocorms under symbiotic germination (SG) and asymbiotic germination (AG) conditions” and more speific description of A and B
Response:L390-391
The attachment was the revised manuscript

Reviewer 2 Report
Comments and Suggestions for Authors
Authors list still contains a mistake:
",_Lin"
Authors wrote that they added, but there is still no line numbers in pdf file to indicate the exact place in which changes must be done.
Numbers of citations are give with or without a space after a word (for example in Abstract and further in the text), please unify the format.
page 1 new sentence should start with capital letter:
"finally. the"
page 2, authors shoud unify the phrase (each time in the middle of a sentence):
"....symbiotic germination" vs "... Symbiotic germination"
page 2
"Gibberellic acids (GAs)" appears two times in the same paragraph
page 3, a space is lacking
"development.These"
kinds of endogenous hormones and their precursors
page 3, errors in English in newly written/corrected parts
"were used to determined" - change to "were used to determine"
"These stages represented seed not germinated, early stage protocorm development and late stage of protocorm development."
change into
"These stages represented not germinated seeds (S0), early stage of protocorm development (S1) and late stage of protocorm development (S2)."
page 3, the lack of a spcae
"ethanol(30%"
section 2.4
"Endogenous plant hormone content of four samples corresponded to two different
germination stage (Stage 1 and Stage 2) of symbiotic and asymbiotic seeds, respectively,
and the free-living mycelium of mycorrhizal fungus and ungerminated seeds (Stage 0)
were"
change into:
Endogenous plant hormone content of four samples corresponding to two different germination stages (S1 and S2) of symbiotic and asymbiotic seeds, respectively, and the free-living mycelium of mycorrhizal fungus and ungerminated seeds (S0) were..."
Section 2.5
Transcriptome analysis of four samples corresponded to two different germination
stage(Stage 1 and Stage 2)of symbiotic and asymbiotic seeds, respectively.
change into
"Transcriptome analysis of four samples corresponding to two different germination stages (S1 and S2) of symbiotic and asymbiotic seeds, respectively, were performed."
rewrite it and avoid repetitions:
"The cleaned reads of D. nobile from the SG and AG libraries were then mapped to the D. nobile genome database, mapping to the genome.
section 2.6
unify the size of font
section 3.2
kinds of endogenous or precursors
change into
page 7
DEGs was already explained at page 4
page 7
authors did not respond to my eearlier suggestion:
"increase font size of KEGG gropu names in Figure 3C and D or their resolution because they seem blurry"
page 8
authors did not respond to my eearlier suggestion:
" add a comma between biosynthesis and transduction if Figure 4 top name"
page 11
authors did not respond to my eearlier suggestion:
"Y axis name of Fig 6d should start with capital letter"
page 12 - missing space
"protocorm(Table 1,"
"expression(Figure 6"
"obvious(Figure 6"
"induced(Figure 6B)."
Author Response
We are very grateful for the constructive comments given by you again. The comments have been very helpful for us to improve our manuscript. We have carefully considered the suggestions, made a massive list of changes. Here are the responses to each point (the revised parts in the text have been marker with red).
Numbers of citations are give with or without a space after a word (for example in Abstract and further in the text), please unify the format.
page 1 new sentence should start with capital letter:
"finally. the"
Response: L56
page 2, authors shoud unify the phrase (each time in the middle of a sentence):
"....symbiotic germination" vs "... Symbiotic germination"
Response: L60
page 2
"Gibberellic acids (GAs)" appears two times in the same paragraph
Response: L61-62
page 3, a space is lacking
"development.These"
Response: It was modified
kinds of endogenous hormones and their precursors
Response:L226
page 3, errors in English in newly written/corrected parts
"were used to determined" - change to "were used to determine"
Response:L122
"These stages represented seed not germinated, early stage protocorm development and late stage of protocorm development."
change into
"These stages represented not germinated seeds (S0), early stage of protocorm development (S1) and late stage of protocorm development (S2)."
Response:L123-124
page 3, the lack of a spcae
"ethanol(30%"
Response: L129
section 2.4
"Endogenous plant hormone content of four samples corresponded to two different
germination stage (Stage 1 and Stage 2) of symbiotic and asymbiotic seeds, respectively,
and the free-living mycelium of mycorrhizal fungus and ungerminated seeds (Stage 0)
were"
change into:
Endogenous plant hormone content of four samples corresponding to two different germination stages (S1 and S2) of symbiotic and asymbiotic seeds, respectively, and the free-living mycelium of mycorrhizal fungus and ungerminated seeds (S0) were..."
Response: L136-139
Section 2.5
Transcriptome analysis of four samples corresponded to two different germination
stage(Stage 1 and Stage 2)of symbiotic and asymbiotic seeds, respectively.
change into
"Transcriptome analysis of four samples corresponding to two different germination stages (S1 and S2) of symbiotic and asymbiotic seeds, respectively, were performed."
Response: L150
rewrite it and avoid repetitions:
"The cleaned reads of D. nobile from the SG and AG libraries were then mapped to the D. nobile genome database, mapping to the genome.
Response:L161
section 2.6
unify the size of font
Response: It was modified
section 3.2
kinds of endogenous or precursors
change into
Response:L226
page 7
DEGs was already explained at page 4
Response: L253
page 7
authors did not respond to my eearlier suggestion:
"increase font size of KEGG gropu names in Figure 3C and D or their resolution because they seem blurry"
Response:It was modified
page 8
authors did not respond to my eearlier suggestion:
" add a comma between biosynthesis and transduction if Figure 4 top name"
Response: It was modified
page 11
authors did not respond to my eearlier suggestion:
"Y axis name of Fig 6d should start with capital letter"
Response: It was modified
page 12 - missing space
"protocorm(Table 1,"
"expression(Figure 6"
"obvious(Figure 6"
"induced(Figure 6B)."
Response:It was modified
The attachment was the revised manuscript
